# Mixing layer transport flux of particulate matter in Beijing, China

Yusi Liu[1], Guiqian Tang[2,3], Libo Zhou[2], Bo Hu[2], Baoxian Liu[4,5], Yunting Li[4,5], Shu Liu[6], Yuesi Wang[2,3,7]

[1] State Key Laboratory of Severe Weather & Key Laboratory for Atmospheric Chemistry of China Meteorology Administration, Chinese Academy of Meteorological Sciences, Beijing 100081, China

[2] State Key Laboratory of Atmospheric Boundary Layer Physics and Atmospheric Chemistry, Institute of Atmospheric Physics, Chinese Academy of Sciences, Beijing 100029, China

[3] Center for Excellence in Urban Atmospheric Environment, Institute of Urban Environment, Chinese Academy of Sciences, Xiamen 361021, China

[4] Beijing Municipal Environmental Monitoring Centre, Beijing 100048, China

[5] Beijing Key Laboratory of Airborne Particulate Matter Monitoring Technology, Beijing 100048, China

[6] Liaoning Provincial Environmental Monitoring & Experiment Center, Shenyang 110031, China

[7] University of Chinese Academy of Sciences, Beijing 100049, China

Correspondence: Guiqian Tang (tgq@dq.cern.ac.cn)

**Abstract**
Quantifying the transport flux (TF) of atmospheric pollutants plays an important role in
understanding the causes of air pollution and in making decisions regarding the prevention and
control of regional air pollution. In this study, the mixing layer height and wind profile were
measured by a ceilometer and Doppler wind radar, respectively, and the characteristics of the
atmospheric dilution capability were analyzed using these two datasets. The ventilation coefficient
(VC) appears to be highest in the spring ($3940 \pm 2110$ $m^2$ $s^{-1}$) and lower in the summer ($2953 \pm 1322$
$m^2$ $s^{-1}$), autumn ($2580 \pm 1601$ $m^2$ $s^{-1}$) and winter ($2913 \pm 3323$ $m^2$ $s^{-1}$). Combined with the
backscatters measured by the ceilometer, vertical profiles of the $PM_{2.5}$ concentration were obtained
and the $PM_{2.5}$ TF in the mixing layer was calculated. The TF was the highest in the spring at $4.33 \pm$
$0.69$ mg $m^{-1}s^{-1}$ and lower in the summer, autumn and winter, when the TF values were $2.27 \pm 0.42$
mg $m^{-1}s^{-1}$, $2.39 \pm 0.45$ mg $m^{-1}s^{-1}$ and $2.89 \pm 0.49$ mg $m^{-1}s^{-1}$, respectively. Air pollutants transport
mainly occurs between 14:00 and 18:00 LT. The TF was large in the pollution transition period
(spring: $5.50 \pm 4.83$ mg $m^{-1}s^{-1}$, summer: $3.94 \pm 2.36$ mg $m^{-1}s^{-1}$, autumn: $3.72 \pm 2.86$ mg $m^{-1}s^{-1}$ and
winter: $4.45 \pm 4.40$ mg $m^{-1}s^{-1}$) and decreased during the heavy pollution period (spring: $4.69 \pm 4.84$
mg $m^{-1}s^{-1}$, summer: $3.39 \pm 1.77$ mg $m^{-1}s^{-1}$, autumn: $3.01 \pm 2.40$ mg $m^{-1}s^{-1}$ and winter: $3.25 \pm 2.77$
mg $m^{-1}s^{-1}$). Our results indicate that the influence of the air pollutants transport in the southern
regions should receive more focus in the transition period of pollution, while local emissions should
receive more focus in the heavy pollution period.
**1. Introduction**
With the rapid development of its economy and industry, as well as its unique local topography,
Beijing has become one of the cities in the world that is most seriously affected by air pollution. As
early as before the 2008 Olympic Games, to fulfill the promise of a "Green Olympics", Beijing's
industries were relocated to surrounding provinces and cities. After the Olympic Games, with the
promulgation of the "Action Plan for Prevention and Control of Air Pollution", Beijing
implemented a series of measures to reduce pollutants, such as raising the emission standards of
motor vehicles and fuel standards for vehicles, changing coal to natural gas, coal to electricity and
so on. These measures have gradually improved Beijing's air quality, with the annual average fine
particulate matter ($PM_{2.5}$) concentration decreasing from 90 $\mu g$ $m^{-3}$ in 2013 to 58 $\mu g$ $m^{-3}$ in 2017
(Cheng et al. 2018a).
Although the Beijing government has been committed in recent years to taking measures that could
ensure a steady improvement in the air quality, there is still great pressure to achieve a continuous
decline in the particulate matter concentration. Beijing is in the north of the North China Plain, with
the south side and the west side the Yanshan Mountains and the Taihang Mountains, respectively.
Affected by the mountains to the northwest, there are more subsiding airflows, a lower mixing layer
height and an extremely limited atmospheric dilution capability. In addition, pollutants tend to
accumulate in front of the mountains due to the influence of southerly winds and the mountain
obstructions. In central and northern China, the increase in $PM_{2.5}$ during the winter is closely related
to the adverse atmospheric dilution conditions (Wang et al. 2016). Therefore, in addition to the
primary emissions and secondary formation, the weak atmospheric dilution capability is also an
important factor leading to the frequent occurrence of serious air pollution in Beijing.
In recent decades, the mixing layer height (MLH) and wind speed (WS) have been two major factors

leading to the annual increases in the aerosol concentration and polluted days during the winter in China (Yang et al. 2016). The low MLH and low WS are also important characteristics of the weak atmospheric dilution capability (Huang et al. 2018; Liu et al. 2018; Song et al. 2014; Tang et al. 2015). The change in the MLH represents the vertical dilution capability of pollutants, and the change in the WS represents the horizontal dilution capability of pollutants. The ventilation coefficient (VC) is usually used to evaluate the vertical and horizontal dilution capability of the atmosphere (Nair et al. 2007; Tang et al. 2015; Zhu et al. 2018). Thus, it is a good choice to use the VC to evaluate the relationship between the atmospheric dilution capability and air pollution in Beijing. Although previous studies have analyzed the relationship between the MLH and pollutants (Geiß et al. 2017; Miao and Liu 2019; Schäfer et al. 2006; Su et al. 2018), studies on the effects of the VC on the particle concentration have been extremely rare.

Although the problem of heavy pollution in northern China has improved in recent years, regional pollution problems remain, especially in the Beijing-Tianjin-Hebei region (Shen et al. 2019). There are three main transport routes affecting Beijing: the northwest path, the southwest path and the southeast path (Chang et al. 2018; Li et al. 2018; Zhang et al. 2018). The occurrence of heavy pollution in Beijing is closely related to the transport of pollutants in the southern regions, mainly in southern Hebei, northern Henan and western Shandong, while the high-speed northwest air mass is conducive to the removal of pollutants from Beijing (Li et al. 2018; Ouyang et al. 2019; Zhang et al. 2018; Zhang et al. 2017). In recent years, the contribution of regional transport to Beijing has been increasing annually, with a trend of 1.2% per year, which reached 31-73% in the summer and 27-59% in the winter (Chang et al. 2018; Cheng et al. 2018b; Wang et al. 2015). High $PM_{2.5}$ concentrations are usually accompanied by high transport flux (TF) within a day in Beijing. As pollution worsens, the contribution of the surrounding areas to the $PM_{2.5}$ in Beijing has risen from 52% to 65% in a month on average in 2016 (Zhang et al. 2018). However, during heavy pollution, the TF decreases in Beijing (Chang et al. 2018; Tang et al. 2015; Zhu et al. 2016).

To solve the regional pollution problem, joint prevention and control have been recommended for a long time. Many studies on regional transport have been carried out, but most observational studies cannot easily quantify the TF due to the lack of particle and wind vertical profiles, and it is still unclear when we need to control the emission sources and in which areas. To solve the above problems, we conducted 2 years of continuous observations on MLH and wind profiles in the Beijing mixing layer and analyzed the mixing layer dilution capability of the atmosphere. Afterwards, using the backscattering coefficient profile, we obtained the vertical $PM_{2.5}$ concentration profiles and calculated the TF profile and mixing layer TF. Finally, using the near-surface $PM_{2.5}$ concentration as an indicator to classify the air pollution degree, we analyzed the TF during the transitional and heavily polluted periods in Beijing and illuminated the main controlling factors.

## 2. Methods

### 2.1 Observational station

To understand the dilution capability characteristics in Beijing, two years of observations were conducted (2016.1.1-2017.12.31). The observational site (BJT) is at the Institute of Atmospheric Physics of the Chinese Academy of Sciences, located west of the Jiande Bridge in the Haidian

District, Beijing (39.98° N, 116.38° W). The north and south sides of the station are the north Third and north Fourth Ring Roads, respectively, and the eastern side is the Beijing-Tibet Expressway. The altitude (a.s.l.) is approximately 60 m. There is no obvious emission source around the observational site except for motor vehicles.

## 2.2 Observations of MLH and wind profiles

To analyze the dilution capability, the MLH was observed by a single-lens ceilometer (CL51, Vaisala, Finland), and the wind profile was simultaneously observed by a Doppler wind radar (Windcube 100s, Leosphere, France).

A single-lens ceilometer measures the attenuated backscatter coefficient profile of the atmosphere by pulsed diode laser lidar technology (910 nm waveband) within a 7.7 km range and determines the MLH through the positions of abrupt changes in the backscattering coefficient profile. In the actual measurement, the measurement interval was 16 s, and the measurement resolution was 10 m. More detailed descriptions are presented in the published literature (Tang et al. 2016; Zhu et al. 2016). In this study, the gradient method (Steyn et al. 1999) is used to determine the MLH; that is, the top of the mixing layer was determined by the maximum negative gradient value in the profile of the atmosphere backscattering coefficient. Moreover, to eliminate the interference of the aerosol layer structure and the detection noise, the MLH was calculated by the improved gradient method after smoothly averaging the profile data (Münkel et al. 2007; Tang et al. 2015).

Doppler wind radar uses the remote sensing method of laser detection and ranging technology and measures the Doppler frequency shift generated by the laser through the backscatter echo signal of particles in the air. The Windcube 100s can provide 3D wind field data within a 3 km range from the system, including u, v and w vectors. In the actual measurement, starting from 100 m, the spatial resolution is 50 m, the WS accuracy is < 0.5 m s$^{-1}$, and the radial WS range is -30 m s$^{-1}$ to 30 m s$^{-1}$.

## 2.3 Other data

During the observations, the hourly PM$_{2.5}$ concentrations of the Beijing Olympic Sports Center (39.99° N, 116.40° W) were obtained from the Ministry of Environmental Protection of China (http://www.zhb.gov.cn/).

## 2.4 Analytical method

The atmospheric dilution is composed of vertical and horizontal dilution, which can be characterized by the MLH and wind speed in the mixing layer (WS$_{ML}$), respectively. The VC (m$^2$ s$^{-1}$) was obtained by combining the MLH (m) and WS$_{ML}$ (m s$^{-1}$) and can be used for a comprehensive evaluation of the vertical and horizontal dilutions. Higher dilution-related parameters (MLH, WS$_{ML}$ and VC) indicate a stronger dilution capability, which is conducive to the transport and dilution of heavy air pollution.

The VC calculation method is as follows:

$$VC = H_{ML} \times WS_{ML}, \tag{1}$$

$$WS_{ML} = \frac{1}{n} \sum_{i=1}^{n} WS_i, \tag{2}$$

$$WS_i = \sqrt{\overline{u_i}^2 + \overline{v_i}^2},$$  (3)
where $WS_{ML}$ is the average WS within the mixing layer, calculated by Eq. (2); $H_{ML}$ is the height of
the mixing layer; $WS_i$ is the WS observed at a certain height, calculated by $u_i$ and $v_i$ in the wind
profile according to Eq. (3); and n is the number of measurement layers in the mixing layer (Nair et
al. 2007).
The TF (mg m$^{-2}$s$^{-1}$) is determined by the WS and the PM$_{2.5}$ concentration in the area under analysis.
The calculation method for a certain height is shown in Eq. (4):
$$TF_{u_i} = u_i \times C_i,$$  (4)
where $C_i$ is the concentration of PM$_{2.5}$ at a certain height. However, it is extremely difficult to
observe the vertical PM$_{2.5}$ concentration in the mixing layer. To obtain the PM$_{2.5}$ concentration
profile, we studied the backscattering coefficient measured by ceilometer, and found that the
concentration of near-surface PM$_{2.5}$ is strongly correlated with the backscattering coefficient at 100
m (Fig. S1). Thus, based on the relationship between the two, the backscattering coefficient profile
can be used to invert the vertical PM$_{2.5}$ concentration profile. Then, the TFs in the mixing layer are
calculated as follows:
$$TF_u = \int_{i=1}^{n} (u_i \times C_i)$$
$$TF_v = \int_{i=1}^{n} (v_i \times C_i)$$  (5)
Through the above method, radial and zonal TFs can be obtained, and vector synthesis in two
directions can be conducted to obtain the main transport direction to find the transport source area.
**3. Results and discussion**
**3.1 Boundary layer meteorology**
**3.1.1 Seasonal variation**
To understand the variations of the atmospheric dilution capability, we carried out continuous
measurements of the MLH and wind profile within the mixing layer over a 2-year period (2016.1.1-
2017.12.31). The availability was verified after MLH elimination by Tang et al. (Tang et al. 2016).
After the exclusion of the data of the MLH under rainy, sandstorm and windy conditions, the data
availability was 95% over the 2-year period, higher than that of previous studies (Mues et al. 2017;
Tang et al. 2016). The availability was the lowest in February, at 86%, and the highest in July, at

166 99%.

In terms of the seasonal variation, the average MLHs for the spring (781 ± 229 m) (value ± standard
deviation) and summer (767 ± 219 m) were higher than those of the autumn (612 ± 166 m) and
winter (584 ± 221 m) (Fig. 1). However, $WS_{ML}$ was different from the MLH in terms of the seasonal
variation, with the largest value 4.6 ± 1.6 m s$^{-1}$ in the spring, followed by the winter (4.1 ± 2.7 m s$^{-1}$)
and autumn (3.7 ± 1.6 m s$^{-1}$), and the smallest value 3.6 ± 1.1 m s$^{-1}$ in the summer. The VC was
calculated by the MLH and wind profile, and the results demonstrate that the dilution capability was
strongest in the spring, as the VC reached as high as 3940 ± 2110 m$^2$ s$^{-1}$. The atmospheric dilution
capabilities for the summer, winter and autumn were similar, with VC values of 2953 ± 1322 m$^2$ s$^{-1}$
$^1$, 2913 ± 3323 m$^2$ s$^{-1}$ and 2580 ± 1601 m$^2$ s$^{-1}$, respectively. A monthly analysis shows that the
atmospheric dilution capability was strongest in May, when the VC was as high as 5161 ± 2085 m$^2$
s$^{-1}$, and worst in December, when the VC was only 1690 ± 1072 m$^2$ s$^{-1}$. The VC value in May was
3.1 times that in December. To analyze the impact of the dilution capacity on PM$_{2.5}$, the seasonal
variation of PM$_{2.5}$ was analyzed. The average PM$_{2.5}$ concentration for the winter (80 ± 87 μg m$^{-3}$)
was the highest, followed by autumn (68 ± 54 μg m$^{-3}$) and spring (67 ± 60 μg m$^{-3}$), and that of the
summer (51 ± 29 μg m$^{-3}$) was the lowest. The lowest monthly average PM$_{2.5}$ concentration was 42
± 26 μg m$^{-3}$ in August. The highest monthly average was in January at 94 ± 100 μg m$^{-3}$, 2.2 times
that in August (Fig. 1).

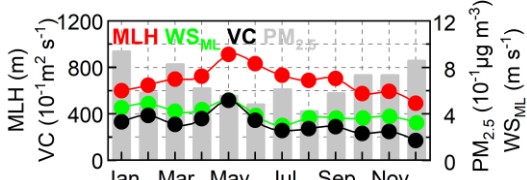

Fig. 1 Monthly variations in mixing layer height (MLH), the wind speed in the mixing layer
(WS$_{ML}$), the ventilation coefficient (VC) and PM$_{2.5}$ in Beijing.
Although there is little difference in the dilution capability between the summer, autumn and winter,
there is serious pollution in the autumn and winter. To analyze this problem, the VC frequency
distribution was studied. The results show that the VC had a high frequency in the range of 1000-
4000 m$^2$ s$^{-1}$ from 2016 to 2017, but the frequency distribution was different in different seasons (Fig.
2). The VC showed a strong dilution capability in the spring, mainly in the range of 2000-5000 m$^2$
s$^{-1}$, with the highest frequency (24%) in the range of 2000-3000 m$^2$ s$^{-1}$. In the summer, the high
frequency of the VC occurred in the range of 1000-4000 m$^2$ s$^{-1}$, which was slightly lower than that
in the spring, and the highest frequency (27%) occurred in the range of 3000-4000 m$^2$ s$^{-1}$.
Additionally, the VC high frequency appeared in lower ranges in the autumn and winter. The VC
occurred at a high frequency of 1000-3000 m$^2$ s$^{-1}$ in the autumn, and the highest frequency occurred
within the range of 2000-3000 m$^2$ s$^{-1}$, accounting for 33%. In the winter, the VC appeared more
frequently in the range of 0-2000 m$^2$ s$^{-1}$ and was the highest in the range of 1000-2000 m$^2$ s$^{-1}$, which
was 28%. In the winter, when the Siberian High transits, strong northwest winds prevail in the
Beijing area (Fig. 5), resulting in the higher frequency of the VC in the range of 1000-2000 m$^2$ s$^{-1}$.
The VC frequency of 0-1000 m$^2$ s$^{-1}$ in the winter was significantly higher than that of the other
seasons, up to 22%, which was 7 times that in the spring, 5 times that in the summer and 2 times
that in the autumn. According to the seasonal variation in the PM$_{2.5}$ concentration, heavy pollution
in the autumn and winter is related to the high frequency of poor atmospheric dilution capability.

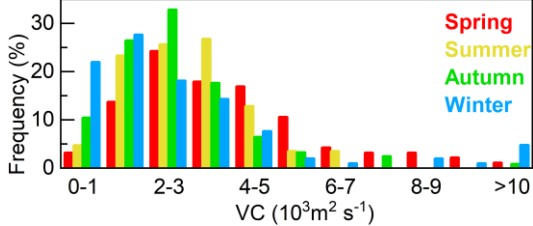

Fig. 2 Frequency distribution of the daily VC from January 2016 to December 2017 in Beijing.

**3.1.2 Diurnal variation**

Moreover, the diurnal variations in the dilution-related parameters during different seasons were analyzed to reveal the diurnal evolution of the atmospheric dilution capability. The peak and trough values of the MLH and VC appeared simultaneously at approximately 15:30 LT and 05:30 LT, respectively. Generally, the daily variation in the MLH is characterized by a low value at night, which increases rapidly after sunrise and reaches the maximum value in the afternoon (Fig. 3a). The daily maximum value of the MLH is seasonal, where it is higher in the spring and summer and lower in the autumn and winter. The daily minimum value of the MLH generally occurs when the mixing layer is stable and is closely related to the WS. The diurnal variation in $WS_{ML}$ is smaller, with a peak at approximately 19:30 LT and a trough at approximately 10:00 LT, which are ~4 h later than the peak and trough of the MLH (Fig. 3b). The diurnal variation in the VC is similar to that of the MLH, showing that the dilution capability is strong before sunset, gradually weakens after sunset and remains stable at night. The dilution capability in the spring was significantly stronger than that during the other seasons, and the maximum daily value reached 8678 $m^2$ $s^{-1}$ (Fig. 3c). The daily maximum values of the VC in the summer, autumn and winter were close, at approximately 5000 $m^2$ $s^{-1}$ (Fig. 3c). The VC growth rate in the spring was significantly higher than that in the other seasons, reaching a maximum at approximately 09:00 LT. In the autumn, the VC growth rate peaked at approximately 10:00 LT, and those in the summer and winter peaked at approximately 11:00 LT. Throughout the year, the VC began to increase during the winter later than in other seasons, at approximately 09:00 LT, indicating that the weaker dilution capability remained for a longer period during the winter. The VC was weakened most rapidly in the spring; however, it was still higher than that of the other seasons after declining. In addition to the spring, the VC in the autumn and winter weakened the most rapidly, and the most slowly in the summer. In general, the vertical and horizontal dilutions are strong in the spring during both the day and night. In the winter, the vertical dilution is weak during the day, and the horizontal dilution during the night is the main component. In the summer, the vertical dilution during the day is dominant.

Notable differences are present when we compare the dilution-related parameters to the $PM_{2.5}$ concentration. The daily maximum $PM_{2.5}$ concentrations in the spring, summer, autumn and winter were 73 μg $m^{-3}$ (11:00 LT), 56 μg $m^{-3}$ (09:00 LT), 78 μg $m^{-3}$ (23:00 LT) and 101 μg $m^{-3}$ (01:00 LT), respectively. The differences between the maximum and minimum were 14 μg $m^{-3}$, 10 μg $m^{-3}$, 20 μg $m^{-3}$ and 38 μg $m^{-3}$, respectively. Thus, the diurnal variation of $PM_{2.5}$ can be divided into two categories: (1) the highest value occurs in the midday in the spring and summer and the overall change is small, and (2) the highest value occurs during the night in the autumn and winter and differs greatly from the lowest value (Fig. 3d). The main causes of air pollution are local emissions and regional transportation. Thus, these results indicate that there are greater local contributions in the autumn and winter and higher regional transport in the spring and summer.

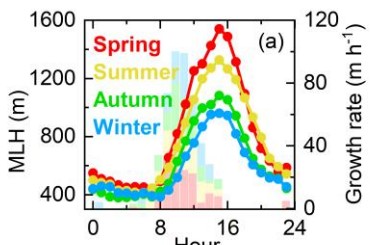 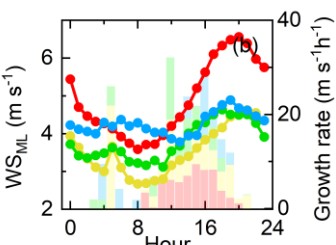

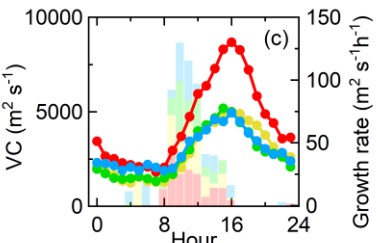 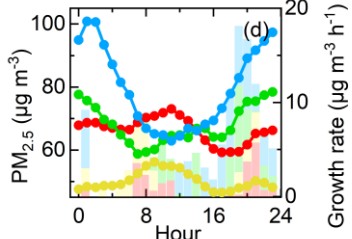

Fig. 3 Diurnal variations and growth rates of MLH (a), $WS_{ML}$ (b), VC (c) and $PM_{2.5}$ (d) in the spring, summer, autumn and winter in Beijing. Diurnal variations are represented by lines and scatters. Growth rates are represented by columns, and only positive values are shown in the figure.

**3.2 Mixing layer TF of $PM_{2.5}$**

**3.2.1 Temporal evolution of TF**

To quantify the transport of $PM_{2.5}$ in Beijing, the transport direction of $PM_{2.5}$ was characterized by the average wind direction in the mixing layer. As shown in Fig. 4, the mixing layer TF in the spring was the largest, reaching $4.33 \pm 0.69$ mg m$^{-1}$s$^{-1}$, and there was no significant difference in the summer, autumn or winter, when the TF values were $2.27 \pm 0.42$ mg m$^{-1}$s$^{-1}$, $2.39 \pm 0.45$ mg m$^{-1}$s$^{-1}$ and $2.89 \pm 0.49$ mg m$^{-1}$s$^{-1}$, respectively. The transport sources of the cold period in Beijing were predominantly from the northwesterly and westerly directions. With temperature warming, the transport direction gradually changed from west to south, mainly southwesterly in the spring and southerly in the summer. The monthly average maximum value of the TF occurred in May, as high as $5.00 \pm 5.21$ mg m$^{-1}$s$^{-1}$ and mainly originated from the southwest direction, accompanied by a strong wind. The minimum value appeared in August, as low as $1.70 \pm 1.73$ mg m$^{-1}$s$^{-1}$, which was mainly transported from western regions, with a small WS. The TF in May was 3 times that in August (Fig. 4). Therefore, the change in the transport direction leads to an obvious seasonal variation in the TF. Overall, the regional transport contributes the most to the $PM_{2.5}$ concentration in the spring, which is mainly related to increased dust activities; regional transport has a smaller contribution in the winter, but there is a high near-surface $PM_{2.5}$ concentration, which indicates that more focus should be given to local emission source control; in the summer and autumn, the southwest airflow transport influence on the Beijing should receive more focus.

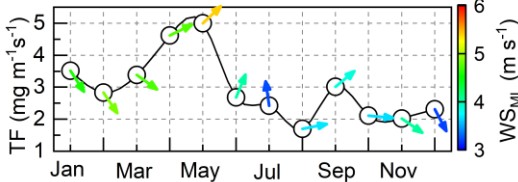

Fig. 4 Seasonal variations in the mixing layer TF of $PM_{2.5}$ and the transport direction.

To understand the regional transport influence on the Beijing area, the diurnal variations of the mixing layer TF were analyzed during different seasons in Beijing. The daily minimum value of the TF appeared at approximately 07:00 LT and was accompanied by a northerly wind. As the average

wind direction in the mixing layer gradually turned south, the daily minimum value of the TF continued to rise until the daily maximum value appeared at approximately 16:00 LT (Fig. 5). Transport mainly occurred between 14:00 and 18:00 LT, which was consistent with the results of a previous study (Ge et al. 2018). In the spring, the WS was the highest, so the peak TF duration was the shortest; it peaked at only 16:00 LT (9.50 mg $m^{-1}s^{-1}$) and then dropped sharply to 1.94 mg $m^{-1}s^{-1}$. Therefore, the diurnal variation in the TF during the spring showed the characteristics of a rapid rise and rapid decline. The peak duration was approximately 3 h for a long time in the summer and autumn, where the daily maximum values were 3.79 mg $m^{-1}s^{-1}$ and 3.63 mg $m^{-1}s^{-1}$ and the minimum values were 1.00 mg $m^{-1}s^{-1}$ and 1.30 mg $m^{-1}s^{-1}$, respectively. The diurnal variation in the TF during the summer and autumn showed the characteristics of a slow rise and slow decline. Specifically, the daily variation had a strong fluctuation in the winter, peaked three times at 14:00 LT (4.06 mg $m^{-1}s^{-1}$), 16:00 LT (4.38 mg $m^{-1}s^{-1}$) and 19:00 LT (4.07 mg $m^{-1}s^{-1}$), then dropped slowly to 1.66 mg $m^{-1}s^{-1}$. Another special point is that in the spring, summer and autumn, a high TF corresponds to a southerly wind, while in the winter, the southerly wind does not appear in the whole transport process; instead, there is a westerly wind, which is influenced by the Siberian High.

Even so, the TF variation patterns can be summarized as that a high TF corresponds to a southerly wind and a low TF corresponds to a northerly wind (Fig. 5). When the average wind direction in the mixing layer changes from north to south, the TF gradually increases from the daily minimum to the daily maximum. The TF increased by 5 times in the spring, 4 times in the summer, and 3 times in the autumn and winter. The current pattern is because areas located in the south of Beijing are heavily polluted and southerly winds help transport pollutants into the city, leading to high TFs in the afternoons (Fig. 5). However, due to the high mixing layer in the spring, the concentration of near-surface $PM_{2.5}$ did not increase. The results further confirm the conclusion that the northwest wind in Beijing is a clean wind (Wang et al. 2015; Zhang et al. 2018). Thus, the northwest wind is conducive to the outward transport of pollutants from Beijing, which helps to alleviate pollution. As a result, there was no high TF in the winter when the northwest wind prevailed. On the other hand, southerly winds are stronger than northerly winds (Fig. 5), which can also result in a high TF. Therefore, the level of the TF is determined by two factors, the WS and $PM_{2.5}$ concentration. In the spring, summer and autumn, a strong south wind prevails in the afternoon. As the south wind is often accompanied by high $PM_{2.5}$ concentrations (Fig. S2), the TF is high. In the winter, the whole day is dominated by westerly and northerly winds. Although the northerly winds are strong, the TF is not high due to the low $PM_{2.5}$ concentration. Generally, a high WS means fast mixing, and the corresponding MLH is also high. At this time, the TF is mainly controlled by the WS. While the WS is low, the mixing speed is slow and the MLH is low. At this time, the TF is mainly controlled by $PM_{2.5}$ concentration. From the above analysis, it can be inferred that if the MLH and WS gradually decrease with the worsening of the pollution, the mixing layer TF is controlled by the WS first and then by the $PM_{2.5}$ concentration, a maximum TF may occur at a critical moment. This moment is neither the moment of the maximum WS nor the moment of the maximum $PM_{2.5}$ concentration but rather should be somewhere in between. This will be discussed in more detail in section 3.3.

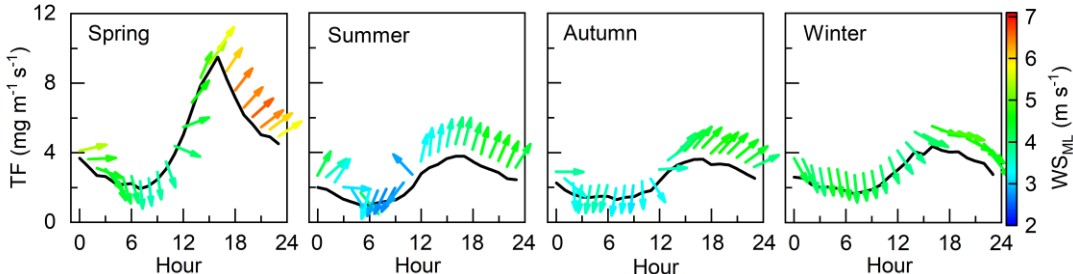

Fig. 5 Diurnal variations in the mixing layer TF of $PM_{2.5}$ and transport direction during different seasons in Beijing.

**3.2.2 Vertical evolution of TF**

After the aforementioned analyses, the transport period is known. To further explore the height of transport, we studied the seasonal variation of the TF profile in combination with the vertical wind and $PM_{2.5}$ profiles. With the increasing altitude, the WS first increases sharply at approximately 200 m and then slowly increases, and the differences between different seasons gradually become significant (Fig. 6a). The WS is always smallest in the summer and strongest in the winter. It is the same in the spring and autumn at 1200 m. Above 1200 m, the WS in the autumn exceeds that in the spring. The $PM_{2.5}$ concentration at 100 m obtained by inversion is highest in the winter (93.7 mg m$^{-2}$s$^{-1}$), similar in the spring and autumn (80.3 mg m$^{-2}$s$^{-1}$ and 75.8 mg m$^{-2}$s$^{-1}$, respectively), and lowest in the summer (53.5 mg m$^{-2}$s$^{-1}$) (Fig. 6b). This finding is consistent with the near-surface results. Below 200 m, the $PM_{2.5}$ concentration is relatively uniform. As the height increases, the $PM_{2.5}$ concentration decreases gradually. Between 200-600 m, the $PM_{2.5}$ concentration begins to decrease rapidly, but the rate of decline is obviously different in different seasons. In the autumn and winter, the reduction rate of the $PM_{2.5}$ concentration is significantly higher than that in the spring and summer. As a result, the spring $PM_{2.5}$ concentration at 400 m begins to be greater than that in the winter; the summer $PM_{2.5}$ concentration at 650 m begins to be greater than that in the autumn and is at the same level as that in the winter. Over 600 m, there is no significant difference in the $PM_{2.5}$ concentration between different seasons, while the WS varies greatly. Therefore, the TF is greatly affected by the WS at high altitudes, and it is greatly influenced by the $PM_{2.5}$ concentration near the ground. The TF in the mixing layer is also affected by the MLH.

The vertical evolution of the TF is different from both the evolution of the WS and $PM_{2.5}$ concentration, and the seasonal variation remains consistent from the near-surface to the upper air. The TF for the spring is the highest, followed by the winter and autumn, and that of the summer is the lowest (Fig. 6c). The vertical variation in the TF increases first and then decreases, and a peak appears at approximately 300 m, with a value of 0.38 mg m$^{-2}$s$^{-1}$ in the spring, 0.19 mg m$^{-2}$s$^{-1}$ in the summer, 0.24 mg m$^{-2}$s$^{-1}$ in the autumn, and 0.31 mg m$^{-2}$s$^{-1}$ in the winter. In the process of the TF lowering, it has different performances in different seasons. In the spring, the decline slows down at approximately 1500 m. The changes in the summer and autumn are similar. After the peak, the TF drops rapidly in the summer and autumn. The decrease rate above 500 m becomes slow, slows down again after 1500 m, and finally the profiles become vertical. In the winter, the TF declines rapidly, followed by fluctuations at approximately 1000 m. The above results preliminarily indicate that the transport mainly occurs within 200-1500 m, which will be evaluated in Sec. 3.3. To sum up,

in the autumn and winter, the high concentration of PM$_{2.5}$ is concentrated near the ground, while the
TF is not large, again indicating that local emissions are the main source of PM$_{2.5}$ in the autumn and
winter; in the spring, affected by high-altitude transport, the PM$_{2.5}$ concentration is high; and in the
summer, both the TF and PM$_{2.5}$ concentration are at their lowest levels, indicating that regional
transport may play an important role in the PM$_{2.5}$ concentration in the summer.

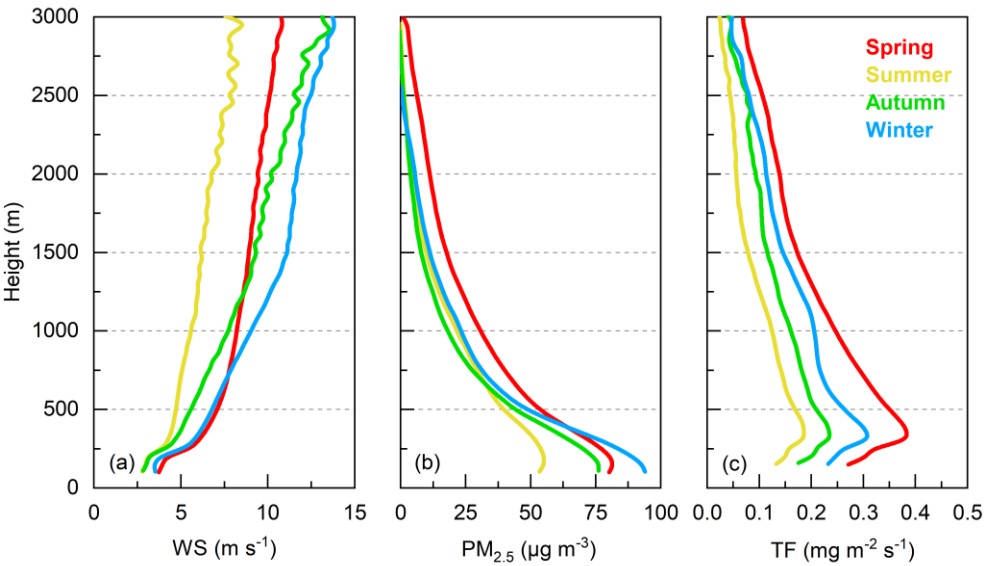

Fig. 6 Vertical profiles of WS (a), PM$_{2.5}$ (b) and TF of PM$_{2.5}$ (c) in different seasons in Beijing.
**3.3 TF under different degrees of air pollution**
Previous studies have demonstrated that transport occurs only in the transition period of pollution,
while it is weak at the peak of pollution (Tang et al. 2015; Zhu et al. 2016). To quantify the transport
impact of different pollution levels, the PM$_{2.5}$ concentration was divided into five levels according
to the "Technical Regulation on Ambient Air Quality Index (on trial)" (HJ 633-2012): PM$_{2.5} \leq$
35 μg m$^{-3}$ (clear days), 35 < PM$_{2.5} \leq$ 75 μg m$^{-3}$ (slight pollution), 75 < PM$_{2.5} \leq$ 115 μg m$^{-3}$ (light
pollution), 115 < PM$_{2.5} \leq$ 150 μg m$^{-3}$ (medium pollution) and PM$_{2.5}$ > 150 μg m$^{-3}$ (heavy pollution).
An interesting phenomenon is that with the increase in altitude, the heavier the pollution near the
ground is, the greater the reduction rate of the PM$_{2.5}$ concentration is (Fig. 7). As a result, there is a
reversal at 1000-1500 m. In other words, the more severe the near-surface pollution, the lower the
high-altitude PM$_{2.5}$ concentration. This is particularly outstanding in the spring: from a clear to a
heavy polluted day, the TF at 100 m was, in turn, 0.15 mg m$^{-2}$s$^{-1}$, 0.26 mg m$^{-2}$s$^{-1}$, 0.32 mg m$^{-2}$s$^{-1}$,
0.39 mg m$^{-2}$s$^{-1}$, 0.66 mg m$^{-2}$s$^{-1}$, and at 2600 m, the values dropped to 0.15 mg m$^{-2}$s$^{-1}$, 0.17 mg m$^{-2}$s$^{-1}$
$^{1}$, 0.13 mg m$^{-2}$s$^{-1}$, 0.10 mg m$^{-2}$s$^{-1}$ and 0.07 mg m$^{-2}$s$^{-1}$, respectively. That is, the lower the pollution
degree, the more vertical the TF tends to be. This is related to the MLH, because a high MLH is
conducive to the diffusion of pollutants in the vertical direction. With the worsening of pollution,
the MLH shows a downward trend (Fig. S3).
According to the previous analysis, two peaks may appear in the TF profile, indicating that the
transport occurs at two different heights, approximately 200 m (low-altitude transport) and 1000 m
(high-altitude transport), respectively. Due to the sudden increase in the WS at approximately 200
m, the low-altitude transport at 200 m is the basic transport height, regardless of the season and the
degree of pollution. In contrast, the high-altitude transport is quite special and mainly occurs in the

winter when there is significant pollution. A small peak of in the TF can also be found on heavy polluted days in the summer. Although the change in the TF profile of medium pollution in the autumn is not as obvious as that in the summer and winter, a small increase can still be seen (Fig. 7). In the case of heavy pollution, the MLH is usually less than 1000 m, while in the case of clear and slight pollution, the MLH is close to the height of high-altitude transport (Fig. S3). Therefore, it can be inferred that the pollutants transported at a high altitude during heavy pollution are stored in the residual layer, and when the mixing layer becomes higher, the pollutants stored in the residual layer diffuse into the mixing layer, affecting the pollution level within the mixing layer. This may be a key contributor to the slight pollution in the summer, autumn and winter, but further research is needed.

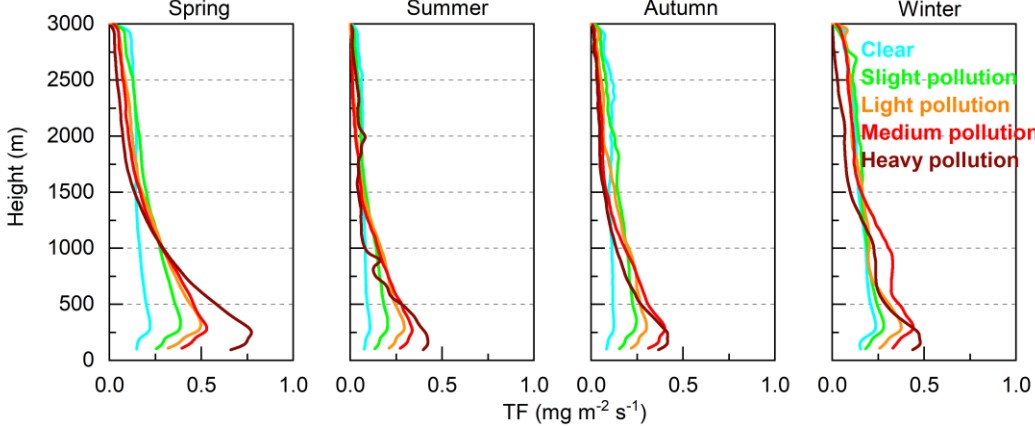

Fig. 7 Vertical profiles of TF of $PM_{2.5}$ under different degrees of pollution in different seasons in Beijing. Clear days: $PM_{2.5} \leq 35$ μg m$^{-3}$, slight pollution: $35 < PM_{2.5} \leq 75$ μg m$^{-3}$, light pollution: $75 < PM_{2.5} \leq 115$ μg m$^{-3}$, medium pollution: $115 < PM_{2.5} \leq 150$ μg m$^{-3}$ and heavy pollution: $PM_{2.5} > 150$ μg m$^{-3}$.

According to the same division method, we further explored the seasonal variation of the TF and transport source in the mixing layer under different pollution degrees. With pollution aggravation, the mixing layer TF in Beijing increased by varying degrees during different seasons, and the transport direction gradually shifted from northwest to south (except during the winter) (Fig. 8). In particular, the mixing layer TF in the spring is significantly higher than that in the other seasons at all pollution degrees, which is 1.1-2.0 times that in the other seasons. This may be caused by the greater amount of dust during the spring. With the pollution deterioration, the TF showed an increasing trend in the initial stage of pollution and a decreasing trend during the heavy pollution period. From medium pollution to heavy pollution, the TF decreased from $5.50 \pm 4.83$ mg m$^{-1}$s$^{-1}$ to $4.69 \pm 4.84$ mg m$^{-1}$s$^{-1}$ in the spring, from $3.94 \pm 2.36$ mg m$^{-1}$s$^{-1}$ to $3.39 \pm 1.77$ mg m$^{-1}$s$^{-1}$ in the summer, from $3.72 \pm 2.86$ mg m$^{-1}$s$^{-1}$ to $3.01 \pm 2.40$ mg m$^{-1}$s$^{-1}$ in the autumn, and from $4.45 \pm 4.40$ mg m$^{-1}$s$^{-1}$ to $3.25 \pm 2.77$ mg m$^{-1}$s$^{-1}$ in the winter. Among them, the largest drop was found in the winter. In the winter, with the pollution aggravation, the transport direction changed from northwest to southwest and finally to the north. In contrast to in the other seasons, the weak north wind was the main wind during heavy pollution in the winter, indicating that regional transport contributed less to the heavy pollution during the winter in Beijing. In the initial stage of pollution, the TF continued to increase, but the rate of increase gradually slowed in the spring and summer. From light pollution to medium pollution, the TF decreased by 0.1mg m$^{-1}$s$^{-1}$ in the spring and increased by only 0.07 mg m$^{-1}$s$^{-1}$ in the summer. It is also not difficult to find from the changes in the TF profile (Fig. 7) that the regional

transport has little impact on the medium pollution in the spring and summer. These results indicate that although the region south of Beijing is the main transport source in Beijing, its contribution is significantly reduced during the severe pollution period. In general, regional transport plays an important role in the initial period of pollution, while local emissions are the main controlling factor during the period of heavy pollution. The parabolic pattern of the TF is the result of a combination of the WS and $PM_{2.5}$ concentration. The TF reaches a threshold during medium pollution, which is the critical moment mentioned above.

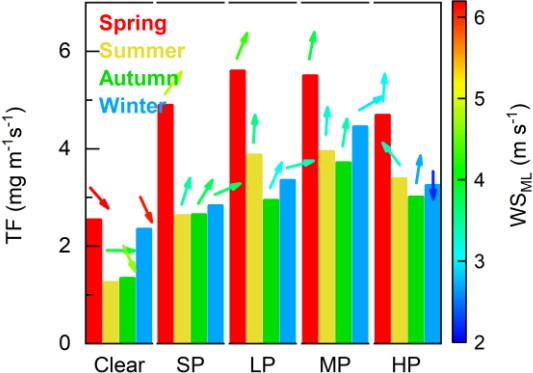

Fig. 8 The mixing layer TF of the $PM_{2.5}$ levels and transport directions under different degrees of pollution in different seasons in Beijing. (SP denotes slight pollution, LP denotes light pollution, MP denotes medium pollution and HP denotes heavy pollution.)

**4. Conclusions**

To understand the characteristics of the $PM_{2.5}$ transport flux in Beijing, the height of the atmospheric mixing layer and the wind profile within the mixing layer in Beijing were observed for a 2-year period. The main conclusions are as follows:

(1) By analyzing the variations in the VC, it is found that the atmospheric dilution capability in Beijing is strongest in the spring and weaker in the summer, autumn and winter. In the spring, the vertical and horizontal dilution capacities are strong; in the autumn and winter, the vertical and horizontal dilution capacities are weak; and in the summer, the vertical dilution capability is strong and the horizontal dilution capability is weak. The diurnal variation in the VC is consistent with the MLH, which shows that the dilution capability is the strongest before sunset, gradually weakens after sunset and remains stable at night. In the spring, the vertical and horizontal dilutions are strong during both day and night. In the winter, the vertical dilution is weak during the day, and the horizontal dilution during the night is the main component. In the summer, the vertical dilution during the day is dominant. Although there is little difference in the diffusivity between the summer, autumn and winter, the poor dilution capability occurs more frequently in the autumn and winter.

(2) The TF is the largest in the spring and smaller in the summer, autumn and winter in Beijing. The high TF mainly comes from southward transport, while the low TF is accompanied by northwest transport. The transport mainly occurred between 14:00 and 18:00 LT, and the height of the transport is at approximately 200 m and 1000 m. Using the $PM_{2.5}$ concentration as a classification index for the air pollution, the results show that the regional transport from the southern area plays an important role in the initial period of pollution, and local emissions are the main controlling factors

in the heavy pollution period, especially in the winter.
To solve the problem of heavy pollution in northern China, joint prevention and control has been
suggested for a long time. Even so, there is still no concrete implementation plan. To break through
this embarrassing situation, this study quantifies TF to explain the time period when the transport
occurs and the main areas affected in Beijing. In this study, the atmospheric dilution capability
during different seasons and the TF during different pollution periods were also discussed. The
important role of transport in the initial period of pollution is emphasized, and local pollutant
emission control is found to be the most effective way of mitigating pollution levels. The research
results are of great significance to the early warning, prevention and control of atmospheric
particulate pollution.
**Data availability**
The data in this study are available from the corresponding author upon request (tgq@dq.cern.ac.cn).
**Author contribution**
GT and YW designed the research, LZ, BH, BL and YunL conducted the measurements. YusL and
GT wrote the paper. SL reviewed and commented on the paper.
**Competing interests**
The authors declare that they have no conflicts of interest to disclose.
**Acknowledgments**
This work was supported by the National Key R&D Program of China (2018YFC0213201), the
Young Talent Project of the Center for Excellence in Regional Atmospheric Environment CAS
(CERAE201802), LAC/CMA (2017A01), the National Research Program for Key Issues in Air
Pollution Control (DQGG0101), the National Natural Science Foundation of China (Nos. 41705113
and 41877312), and the Foundation of the Chinese Academy of Meteorological Sciences
(2019Y001).

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
