# Peer review of "Mixing layer transport flux of particulate matter in Beijing, China"

_Atmospheric Chemistry and Physics, 2019_

## Referee Comment (RC1) · Anonymous Referee #1 · 27 Feb 2019

To quantifying the transport flux of atmospheric pollutants for understanding the causes of atmospheric pollution levels and development of decisions regarding the prevention and control of atmospheric pollution, the mixing layer height and wind profile inside the mixing layer were measured by ceilometer and doppler wind radar, respectively. The variation characteristics of atmospheric transport capacity (TC) were analyzed on this data base: TC is strongest in spring and weakest in autumn. The TC influence on the PM2.5 concentration was determined and there shows a strong inverse correlation between the PM2.5 and TC in spring, autumn and winter and a weak positive correlation in summer. The transport flux (TF) of fine particles in Beijing is highest in spring and lower in the other three seasons. The transport occurs mainly between 14:00 and 18:00 LT. The TF was large in the pollution transition period and decreased during

heavy pollution periods. General comments The application of TC, TF and VC should be explained in more detail: why these parameters are used and which advantages it provides in comparison to alternative parameters. It is concluded that the transportation influence in southern regions is of higher influence in the transition period of pollution, while local emissions are more important in the heavy pollution period. My main concern is why the whole discussion with TC, TF and VC up to chapter 3.2 is without wind direction. In chapter 3.3 it would be helpful to discuss MLH also. The conclusions are a summary and in this summary no relation to the existing knowledge / papers are given. What is new and what is supported by this study? The paper addresses relevant scientific tasks. The paper presents novel concepts, ideas and tools. The scientific methods and assumptions are valid and clearly outlined so that substantial conclusions are reached. The description of experiments and calculations allow their reproduction by fellow scientists. The quality of the figures is good. The figure captions should be improved so that these are understandable without the overall manuscript: terms must be explained, description of parameters. The related work is well cited so that the authors give proper credit to related work and own new contribution. The title as well as the abstract reflects the whole content of the paper. The overall presentation is well structured and clear. The language is fluent but must be improved in much details. The mathematical formulae, symbols, abbreviations, and units are generally correctly defined and used. Specific Comments Line 46: The values are valid for which time period? Line 57: How TC is defined? Reference? Line 59: What about wind direction? Line 64: How VC is defined? Reference? Line 81: When this happened? Lines 110 – 113: This explanation is not correct. Explain clearly what do you mean. Line 116: What is $-d\beta/dx$? Line 128: time resolution not time accuracy Lines 142 – 144: Why this is an explanation? Height profile instead of "by height" Line 353: How PM2.5 concentration is related to photochemical reactions? Line 366: concentration column? What do you mean? Technical corrections Indicate if there are papers in Chinese.

---

## Referee Comment (RC2) · Anonymous Referee #2 · 5 Mar 2019

The current study explores the seasonal source of PM2.5 pollution in Beijing by quantifying the transport flux based on measurements of mixing layer height and wind profile. In particular, this study raises two questions that are rarely addressed in previous studies: (1) effects of ventilation coefficient on PM2.5, and (2) observational quantification of transport fluxes. This topic is of broad interest to both the scientific community and policy-makers. The datasets analyzed in the study is valuable. However, the current analyses do not clearly address the questions raised in the beginning. In addition, the data and method section requires some clarification. Therefore, I recommend major revision.

Specific comments: 1. I suggest changing the second question to emphasize its scientific merit. By quantifying transport fluxes from observation, what scientific question do

you want to address? 2. Section 2.2 describe the method to determine MLH. Although details are provided in earlier papers, necessary steps should be clearly mentioned in the current paper, e.g. line 113-115 averaging the profile over time? If so, over what time window, daily, hourly? 3. Section 2.4 cited a previous study to support the assumption that backscattering coefficient is relatively uniform in the mixing layer. I think your ceilometer observations include backscatter profile. Does your data quantitatively support this assumption? 4. On line 156-158 and following statements, what is the number behind the ïĆś sign? 5. I suggest using the same color scheme for each season in Fig. 2 and Fig. 3. 6. Why didn't you show diurnal variations and growth rates of PM2.5 in Fig. 2? It seems directly relevant to the first scientific question. 7. In Fig.3, it is worth discussing higher frequency of high VC (> 103 m2 s-1) in winter, is it due to high wind speed associated with frontal passage? 8. In Fig.4, it seems to me that the dominant southerly wind partly explains the positive correlation between wind speed and PM2.5 in summer. 9. I don't think the conclusion on lines 289-294 that southerly wind is "dirtier" directly comes from Figure 5 and 6 Flux variation comes from PM2.5 and wind speed, it could be that southerly wind are generally stronger. In order to demonstrate this point, it will help to add PM2.5 fields in Figure 5 and Figure 6. Another way to demonstrate this conclusion is to show wind rose and flux rose, and PM2.5 composite in different wind directions.

---

## Referee Comment (RC3) · Anonymous Referee #3 · 7 Mar 2019

General Comments:

The manuscript presents a good investigation by studying the transport flux of particulate matter in the mixing layer over Beijing area, one of the heavily polluted places in the country. The study employs ceilometer, Doppler wind radar, and other meteorological measurement techniques to determine the transport flux in the region.

Overall, the manuscript constitutes a good research article with clear conclusions, high quality figures, and great organization of the data. However, there seems to be a lot of room for English language improvement.

Specific Comments:

1, Line 26, define "fine particle" for its first appearance, e.g., PM2.5 or something else.

[Figure]

2, Line 31, recommend changing to "Transport mainly occurs between 14:00 and 18:00 LT".

3, Line 41, recommend changing "other provinces and cities" to "surrounding provinces and cities"

4, Line 46, define fine particulate matter as PM2.5 also if it is what he authors mean

5, Line 49, recommend changing "a steady decrease in poor air quality" to "steady improvement in air quality"

6, Line 77, recommend changing "...1.2% yr-1..." to "1.2 percent per year"

7, Line 86, recommend changing "...the reliability of the model will decrease" to "...the reliability of the model cannot be guaranteed"

8, Line 91, recommend organizing it as "...transport flux (TF) in the mixing layer..."

9, Line 156-158, the way this sentence and next one were constructed will really confuse the readers. "Seasonal variation" means and focuses on the variation, i.e, the standard deviation. I think the authors is trying to express something like this: "In terms of seasonal variation, the means of MLH for spring and summer are relatively higher than those of fall/autumn and winter. However, WS was quite different from MLH, ...". For Line 166-169, according adjustment is recommended for the discussion of PM2.5 to avoid confusion.

10, Line 163-164, recommend changing to "...The average TC for summer, winter, and autumn were quite similar, with the VC values...."

11, Line 233-234, does the authors want to express this: "When MLH, WSML and VC were lower than 400 m, 2.5 m s-1 and 1500 m2 s-1, respectively, the PM2.5 concentration decline sharply with these parameters increasing"? It is hard to imagine air pollution declines at these conditions not in favor of atmospheric dispersion.

12, Line 261, I think May TF of 269 mg m-1 s-1 was 1.5 times higher than August TF of

106 mg m-1 s-1. Alternatively, you can express it as "May TF was 2.5 times of August TF".

13, A general comment: when using "transport" and "transportation", try to clarify it and avoid the ambiguity by meaning the transportation sector like vehicle emissions, since it is also great contributing factor for fine particle concentration.

14, Line 361-364, the expression in this segment could be revised to avoid negative image of the conclusion.

Technical corrections:

1, Line 20, change "atmospheric pollution" to "air pollution"

2, Line 24, change "weakens" to "weaker" or make alternative grammar corrections

3, Line 35, change "transportation influence" to "influence/impact of (air pollutants) transport", otherwise it seems to mean the influence of transportation section like vehicles

4, Line 45, change "the Beijing's air quality" to "Beijing's air quality"

5, Line 48, change "Although Beijing's government has been dedicated. . ." to "Although Beijing government has dedicated. . ."

6, Line 49-50, change ". . .ensure the continuous decline. . ." to ". . .ensure continuous decline. . ." or ". . .ensure the continued decline. . ."

7, Line 109, change ". . .More detail descriptions. . ." to "More detailed descriptions. . ."

8, Line 116, change". . .remote sensor method. . ." to "remote sensing method. . ."

9, Line 120, change the long dash to short dash or change it to "to"

10, Line 150, change ". . .we carried out continuously measured. . ." to ". . .we continuously measured. . ." or "we carried out continuous measurement of. . ."

Line 184, change "stable" to "relatively smaller"

Line 185, recommend changing to "which are ∼4 h later than the peak and trough of MLH..."

Line 193, change "at the latest" to "later than other seasons". "At the latest" means something else like a deadline.

Line 195, change "TC" to "VC" or change "VC" to "TC", so that the same parameter is compared, even though we VC is used to express the magnitude of TC.

Line 236, change to "...than other seasons..."

Line 243, change "indicator factors" indicators" or "indicating factors"

Line 255-256, need improvement for this expression: "The northwesterly and westerly directions were the main transport sources of the cold period in Beijing."

Line 257, change "increased" to "changed"

Line 286, change "rules" to "patterns"

Line 297, change "4" to "four", please refer to manuscript preparation guidance about numbers.

Line 299, recommend changing "and we must pay attention to local pollutant emission control" to "and local pollutant emission control is the most effective way of mitigating pollution levels"

Line 346-347, change "the concentration of pollutants has a good relationship with VC" to "the concentration of pollutants is significantly correlated with VC"

---

## Author Comment (AC2) · 10 Jun 2019

We would like to thank you for your comments and helpful suggestions. We revised our manuscript accordingly. General Comments: The current study explores the seasonal source of PM2.5 pollution in Beijing by quantifying the transport flux based on measurements of mixing layer height and wind profile. In particular, this study raises two questions that are rarely addressed in previous studies: (1) effects of ventilation coefficient on PM2.5, and (2) observational quantification of transport fluxes. This topic is of broad interest to both the scientific community and policy-makers. The datasets analyzed in the study is valuable. However, the current analyses do not clearly address the questions raised in the beginning. In addition, the data and method section require some clarification. Therefore, I recommend major revision. Specific Comments: Comment 1:

[Figure]

I suggest changing the second question to emphasize its scientific merit. By quantifying transport fluxes from observation, what scientific question do you want to address? Response 1: Thank you for your helpful suggestion. We have emphasized the scientific merit of the second question and added it to the introduction, as follows: Although the problem of heavy pollution in northern China has improved in recent years, regional pollution problems remain, especially in the Beijing-Tianjin-Hebei region (Shen et al. 2019). To solve the regional pollution problem, joint prevention and control have been recommended for a long time. Many studies on regional transport have been carried out, but most observational studies cannot easily quantify the transport flux due to the lack of particle and wind vertical profiles, and it is still unclear when we need to control the emission sources and in which areas. In this study, we used the backscattering coefficient measured by a ceilometer and wind profile to quantify the transport fluxes to solve the problems mentioned above. Comment 2: Section 2.2 describe the method to determine MLH. Although details are provided in earlier papers, necessary steps should be clearly mentioned in the current paper, e.g. line 113-115 averaging the profile over time? If so, over what time window, daily, hourly? Response 2: Thank you for your helpful suggestion. The text has been revised to "the MLH was calculated by the improved gradient method after smoothly averaging the profile data". More details are as follows: Because the lifetime of the particles can be several days or even weeks, the distribution of the particle concentration in the MLH is more uniform than that of the gaseous pollution. However, the particle concentration in the mixing layer and that in the free atmosphere are significantly different. In the attenuated backscatter coefficient profile, the position at which a sudden change occurs in the profile indicates the top of the atmospheric mixing layer. In this study, we used the Vaisala software product BL-VIEW to determine the MLH. The time averaging is dependent on the current signal noise. Height averaging intervals range from 80 m at ground level to 360 m at a 1600 m height and beyond. Additional features of this algorithm, which is used in the Vaisala software product BL-VIEW, include cloud and precipitation filtering and outlier removal. Because the aerosol concentrations are particularly low above the BLH and the BLH

in the Beijing area is usually lower than 4 km, we halved the detection range to 7.7 km to reinforce the echo signals and reduce the detection noise. Comment 3: Section 2.4 cited a previous study to support the assumption that backscattering coefficient is relatively uniform in the mixing layer. I think your ceilometer observations include backscatter profile. Does your data quantitatively support this assumption? Response 3: Thank you for your helpful suggestion. Although previous studies have shown that the concentration of particulate matter in the mixing layer is basically uniform, there are still large differences in some time periods, especially in the time periods with transport effects. Based on your suggestions and those of Reviewer 2, we find it inappropriate to so rashly use the near-surface PM2.5 concentration as the concentration in the mixing layer. Because the ceilometer can measure the atmospheric backscattering coefficient, it is possible to obtain the vertical profile of the particles. Therefore, in the revised draft, we analyzed the relationship between the backscattering coefficient at 100 m measured by ceilometer and the near-surface PM2.5 concentration, discussed their correlations in different seasons, and obtained the fitting curves of different seasons. Using these four equations, we obtained the PM2.5 concentration at different heights in different seasons. According to this result, we have recalculated the TF in the revised draft. Comment 4: On line 156-158 and following statements, what is the number behind the ï′C′s sign? Response 4: Thank you for your helpful suggestion. I guess you mean "±". The number after the "±" represents the standard deviation, a measure of the dispersion of the data. An explanation has been added where the notation first appeared. Comment 5: I suggest using the same color scheme for each season in Fig. 2 and Fig. 3. Response 5: Thank you for your helpful suggestion. The color scheme has been unified. Comment 6: Why didn't you show diurnal variations and growth rates of PM2.5 in Fig. 2? It seems directly relevant to the first scientific question. Response 6: Thank you for your helpful suggestion. The diurnal variations of the PM2.5 and the corresponding analysis have been added. More details are as follows: Notable differences are present when we compare the dilution-related parameters to PM2.5. The daily maximum PM2.5 concentrations in the spring, summer, autumn and

winter were 73 $\mu$g m-3 (11:00 LT), 56 $\mu$g m-3 (09:00 LT), 78 $\mu$g m-3 (23:00 LT) and 101 $\mu$g m-3 (01:00 LT), respectively. The differences between the maximum and minimum were 14 $\mu$g m-3, 10 $\mu$g m-3, 20 $\mu$g m-3 and 38 $\mu$g m-3, respectively. Thus, the diurnal variation of PM2.5 can be divided into two categories: (1) the highest value occurs in the midday in the spring and summer and the overall change is small and (2) the highest value occurs during the night in the autumn and winter and differs greatly from the lowest value (Fig. 1). The main causes of air pollution are local emissions and regional transport. Thus, these results indicate that there is a greater local contribution in the autumn and winter and higher regional transport in the spring and summer. Comment 7: In Fig.3, it is worth discussing higher frequency of high VC (> 103 m2 s-1) in winter, is it due to high wind speed associated with frontal passage? Response 7: Thank you for your helpful suggestion. We agree with you. In winter, when the Siberian High transits, strong northwest winds prevail in the Beijing area (Fig. 2), resulting the higher frequency of the VC in the range of 1000-2000 m2 s-1. We explained this point in section 3.1.1 of the revised draft. Comment 8: In Fig.4, it seems to me that the dominant southerly wind partly explains the positive correlation between wind speed and PM2.5 in summer. Response 8: Thank you for your helpful suggestion. The southern wind generally appeared at 12:00-2:00 LT, and the high PM2.5 concentration generally appeared at 6:00-13:00 LT; therefore, there was no significant relationship between the two. In addition, due to the improper discussion of this section in the original text, we have deleted this section to avoid confusion. Comment 9: I don't think the conclusion on lines 289-294 that southerly wind is "dirtier" directly comes from Figure 5 and 6 Flux variation comes from PM2.5 and wind speed, it could be that southerly wind are generally stronger. In order to demonstrate this point, it will help to add PM2.5 fields in Figure 5 and Figure 6. Another way to demonstrate this conclusion is to show wind rose and flux rose, and PM2.5 composite in different wind directions. Response 9: Thank you for your helpful suggestion. According your suggestion, the diurnal variation of the PM2.5 concentration and the wind radar were added, and we found that the level of the TF is determined by two factors, the WS and PM2.5 concentration. In the spring, summer

and autumn, the strong south wind prevails in the afternoon. As the south wind is often accompanied by a high PM2.5 concentration (Fig. 3), the TF is high. In the winter, the whole day is dominated by westerly and northerly winds. Although the northerly winds are strong, the TF is not high due to the low PM2.5 concentration. Generally, a high WS means fast mixing, and the corresponding MLH is also high. At this time, the TF is mainly controlled by the WS. When the WS is low, the mixing speed is slow, and the MLH is low. At this time, the TF is mainly controlled by the PM2.5 concentration. From the above analysis, it can be inferred that if the MLH and WS gradually decrease with the worsening of the pollution, the mixing layer TF is controlled by the WS first and then by the PM2.5 concentration, and the maximum TF may occur at a critical moment. This moment is neither the moment of the maximum WS nor the moment of the maximum PM2.5 concentration but should be somewhere in between. References: Shen, Y., L. Zhang, X. Fang, H. Ji, X. Li, and Z. Zhao: Spatiotemporal patterns of recent PM2.5 concentrations over typical urban agglomerations in China, Sci Total Environ, 655, 13-26, https://doi.org/10.1016/j.scitotenv.2018.11.105, 2019.
* * *
[Figure]

[Figure]

**Fig. 1.** Diurnal variations and growth rates of the MLH (a), WSML (b), VC (c) and PM2.5 (d) in the spring, summer, autumn and winter in Beijing. Diurnal variations are represented by lines and scatters.

[Figure]

**Fig. 2.** Diurnal variations in the mixing layer transport flux of PM2.5 and transport direction during different seasons in Beijing.

[Figure]

**Fig. 3.** The wind radar in different seasons in Beijing.

---

## Author Response (AR1)

Dear Editors and Reviewers:

Thank you all for your comments concerning our manuscript entitled "Mixing layer
transport flux of particulate matter in Beijing, China" (Manuscript ID: acp-2019-141).
The comments were all valuable and very helpful for revising and improving the
manuscript and provided important guiding significance for our research. We have
studied the comments carefully and made corrections that we hope will be met with
approval. The revised parts of the manuscript are shown using the "Track Changes"
feature in Word. Below, we have provided the reviewers' comments for ease of reading
and have added our response after each comment.

List of Responses

**Responses to the comments from Reviewer #1**

We would like to thank you for your comments and helpful suggestions. We have
revised the manuscript accordingly.

**General Comments:**
To quantifying the transport flux of atmospheric pollutants for understanding the causes
of atmospheric pollution levels and development of decisions regarding the prevention
and control of atmospheric pollution, the mixing layer height and wind profile inside
the mixing layer were measured by ceilometer and doppler wind radar, respectively.
The variation characteristics of atmospheric transport capacity (TC) were analyzed on
this data base: TC is strongest in spring and weakest in autumn. The TC influence on
the $PM_{2.5}$ concentration was determined and there shows a strong inverse correlation
between the $PM_{2.5}$ and TC in spring, autumn and winter and a weak positive correlation
in summer. The transport flux (TF) of fine particles in Beijing is highest in spring and
lower in the other three seasons. The transport occurs mainly between 14:00 and 18:00
LT. The TF was large in the pollution transition period and decreased during heavy
pollution periods.

**Comment 1:**
The application of TC, TF and VC should be explained in more detail: why these
parameters are used and which advantages it provides in comparison to alternative
parameters.
**Response 1:**
Thank you for your helpful suggestion. After careful consideration, we think that
"atmospheric transport capacity" is prone to ambiguity, so we changed this term to
"atmospheric dilution capability". Atmospheric dilution is composed of vertical and
horizontal dilutions, which can be characterized by the mixing layer height (MLH) and
wind speed in the mixing layer ($WS_{ML}$), respectively. The ventilation coefficient (VC)
is obtained by combining MLH and $WS_{ML}$ and can be used for a comprehensive
evaluation of the vertical and horizontal dilutions, where a higher VC indicates a
stronger dilution capability. The TF represents the transport flux of $PM_{2.5}$, which can quantify the amount of pollutants passing through the area to assess the impact of
regional transport. To avoid confusion, changes were made in the paper.

**Comment 2:**

It is concluded that the transportation influence in southern regions is of higher
influence in the transition period of pollution, while local emissions are more important
in the heavy pollution period. My main concern is why the whole discussion with TC,
TF and VC up to section 3.2 is without wind direction. In section 3.3 it would be helpful
to discuss MLH also.

**Response 2:**

Thank you for your helpful suggestion. After careful consideration, we have revised the
structure of the paper according to your suggestion. Section 3.1 mainly discusses the
seasonal and diurnal variations of the atmospheric dilution capability and $PM_{2.5}$
concentration; section 3.2 mainly discusses the evolution of the TF, both temporally
and spatially; and section 3.3 analyzes the evolution of the TF under different pollution
degrees in detail. The revised structure will make it easier for readers to understand.
Thank you very much for your suggestions.

In addition, we have added the evolution of the MLH under different pollution degrees
in section 3.3 as suggested. We found that the MLH decreases gradually with the
worsening of the pollution (Fig. 1). This result also supports the conclusion that the
transport is weak during heavy pollution.

[Figure]

Fig. 1 Mixing layer height under different degrees of pollution in different seasons in Beijing.

**Comment 3:**

The conclusions are a summary and in this summary no relation to the existing knowledge / papers are given. What is new and what is supported by this study? The paper addresses relevant scientific tasks. The paper presents novel concepts, ideas and tools. The scientific methods and assumptions are valid and clearly outlined so that substantial conclusions are reached. The description of experiments and calculations allow their reproduction by fellow scientists.

**Response 3:**

Thank you for your helpful suggestion. Joint prevention and control have been recommended for a long time to solve the problem of heavy pollution in northern China. Even so, no concrete implementation plan has been established. To break through this embarrassing situation, this study quantifies the transport flux to explain the time period when the transport occurs, the main areas affected in Beijing and the height of transport. The important role of transport in the initial period of pollution is emphasized. The innovation of this study has been added to the conclusion.

**Comment 4:**

The quality of the figures is good. The figure captions should be improved so that these are understandable without the overall manuscript: terms must be explained, description of parameters.

**Response 4:**

Thank you for your helpful suggestion. According your suggestion, we added more detail to make the figures more readable, such as descriptions of the parameters and explanations of the abbreviations.

**Specific Comments:**

**Comment 1:**

Line 46: The values are valid for which time period?

**Response 1:**

Thank you for your helpful suggestion. The phrase has been revised to "the annual average fine particulate matter concentration".

**Comment 2:**

Line 57: How TC is defined? Reference? Line 59: What about wind direction? Line 64: How VC is defined? Reference?

**Response 2:**

Thank you for your helpful suggestion. As mentioned in the response to comment 1 in the "General Comments", we changed "TC" to "atmospheric dilution capability". Definitions of the atmospheric dilution capability and VC have also been described in the beginning of section 2.4. The wind direction in this study refers to the average wind direction in the mixing layer. For ease of understanding, we modified the expression to "average wind direction in the mixing layer".

**Comment 3:**

Line 81: When this happened?

**Response 3:**

Thank you for your helpful suggestion. This event happened in 2016, and this information has been added to the paper.

**Comment 4:**

Lines 110 – 113: This explanation is not correct. Explain clearly what do you mean.

**Response 4:**

Thank you for your helpful suggestion. This section was removed during the revision process.

**Comment 5:**

Line 116: What is $-(d\beta/dx)$?

**Response 5:**

Thank you for your helpful suggestion. $\beta$ is the backscatter coefficient, and x is the distance between the lidar and scattering volume (Münkel et al. 2007). $-(d\beta/dx)$ represents the maximum negative gradient value in this paper. Considering that $-(d\beta/dx)$ has no practical meaning in the paper, it has been deleted.

**Comment 6:**

Line 128: time resolution not time accuracy

**Response 6:**

Thank you for your helpful suggestion. This section was corrected the revision process. The phrase "A time accuracy of 1 h" has been revised to "hourly".

**Comment 7:**

Lines 142 – 144: Why this is an explanation? Height profile instead of "by height"

**Response 7:**

Thank you for your helpful suggestion. Although previous studies have shown that the concentration of particulate matter in the mixing layer is basically uniform, there are still large differences in some time periods, especially in time periods with transport effects. Based on your suggestion and that of Reviewer 2, we find it inappropriate to so rashly use the near-surface $PM_{2.5}$ concentration as the concentration in the mixing layer. Because the ceilometer can measure the atmospheric backscattering coefficient, it is possible to obtain the vertical profile of the particles. Therefore, in the revised draft, we analyzed the relationship between the backscattering coefficient at 100 m measured by the ceilometer and the near-surface $PM_{2.5}$ concentration, discussed their correlations in different seasons, and obtained the fitting curves of different seasons. Using these four equations, we obtained the $PM_{2.5}$ concentration at different heights in different seasons. According to this result, we have recalculated the TF in the revised draft.

**Comment 8:**

Line 353: How $PM_{2.5}$ concentration is related to photochemical reactions?

**Response 8:**

Thank you for your helpful suggestion. Through subsequent analysis, we found that our
previous inference was wrong. Considering that this part is not closely related to the
topic, it has been deleted from the manuscript.

**Comment 9:**
Line 366: concentration column? What do you mean? Technical corrections Indicate if
there are papers in Chinese.
**Response 9:**
Thank you for your helpful suggestion. We apologize for this mistake. We have revised
"concentration column" to "column concentration".

**Responses to the comments from Reviewer #2**

We would like to thank you for your comments and helpful suggestions. We revised
our manuscript accordingly.

**General Comments:**
The current study explores the seasonal source of $PM_{2.5}$ pollution in Beijing by
quantifying the transport flux based on measurements of mixing layer height and wind
profile. In particular, this study raises two questions that are rarely addressed in
previous studies: (1) effects of ventilation coefficient on $PM_{2.5}$, and (2) observational
quantification of transport fluxes. This topic is of broad interest to both the scientific
community and policy-makers. The datasets analyzed in the study is valuable. However,
the current analyses do not clearly address the questions raised in the beginning. In
addition, the data and method section require some clarification. Therefore, I
recommend major revision.

**Specific Comments:**
**Comment 1:**
I suggest changing the second question to emphasize its scientific merit. By quantifying
transport fluxes from observation, what scientific question do you want to address?
**Response 1:**
Thank you for your helpful suggestion. We have emphasized the scientific merit of the
second question and added it to the introduction, as follows:
Although the problem of heavy pollution in northern China has improved in recent
years, regional pollution problems remain, especially in the Beijing-Tianjin-Hebei
region (Shen et al. 2019). To solve the regional pollution problem, joint prevention and
control have been recommended for a long time. Many studies on regional transport
have been carried out, but most observational studies cannot easily quantify the
transport flux due to the lack of particle and wind vertical profiles, and it is still unclear
when we need to control the emission sources and in which areas. In this study, we used
the backscattering coefficient measured by a ceilometer and wind profile to quantify
the transport fluxes to solve the problems mentioned above.

**Comment 2:**
Section 2.2 describe the method to determine MLH. Although details are provided in
earlier papers, necessary steps should be clearly mentioned in the current paper, e.g.
line 113-115 averaging the profile over time? If so, over what time window, daily,
hourly?
**Response 2:**
Thank you for your helpful suggestion. The text has been revised to "the MLH was
calculated by the improved gradient method after smoothly averaging the profile data".
More details are as follows:
Because the lifetime of the particles can be several days or even weeks, the distribution
of the particle concentration in the MLH is more uniform than that of the gaseous
pollution. However, the particle concentration in the mixing layer and that in the free
atmosphere are significantly different. In the attenuated backscatter coefficient profile,
the position at which a sudden change occurs in the profile indicates the top of the
atmospheric mixing layer. In this study, we used the Vaisala software product BL-
VIEW to determine the MLH. The time averaging is dependent on the current signal
noise. Height averaging intervals range from 80 m at ground level to 360 m at a 1600
m height and beyond. Additional features of this algorithm, which is used in the Vaisala
software product BL-VIEW, include cloud and precipitation filtering and outlier
removal. Because the aerosol concentrations are particularly low above the BLH and
the BLH in the Beijing area is usually lower than 4 km, we halved the detection range
to 7.7 km to reinforce the echo signals and reduce the detection noise.

**Comment 3:**
Section 2.4 cited a previous study to support the assumption that backscattering
coefficient is relatively uniform in the mixing layer. I think your ceilometer
observations include backscatter profile. Does your data quantitatively support this
assumption?
**Response 3:**
Thank you for your helpful suggestion. Although previous studies have shown that the
concentration of particulate matter in the mixing layer is basically uniform, there are
still large differences in some time periods, especially in the time periods with transport
effects. Based on your suggestions and those of Reviewer 2, we find it inappropriate to
so rashly use the near-surface $PM_{2.5}$ concentration as the concentration in the mixing
layer. Because the ceilometer can measure the atmospheric backscattering coefficient,
it is possible to obtain the vertical profile of the particles. Therefore, in the revised draft,
we analyzed the relationship between the backscattering coefficient at 100 m measured
by ceilometer and the near-surface $PM_{2.5}$ concentration, discussed their correlations in
different seasons, and obtained the fitting curves of different seasons. Using these four
equations, we obtained the $PM_{2.5}$ concentration at different heights in different seasons.
According to this result, we have recalculated the TF in the revised draft.

**Comment 4:**
On line 156-158 and following statements, what is the number behind the ï´C´s sign?

 **Response 4:**

Thank you for your helpful suggestion. I guess you mean "±". The number after the "±" represents the standard deviation, a measure of the dispersion of the data. An explanation has been added where the notation first appeared.

**Comment 5:**

I suggest using the same color scheme for each season in Fig. 2 and Fig. 3.

**Response 5:**

Thank you for your helpful suggestion. The color scheme has been unified.

**Comment 6:**

Why didn't you show diurnal variations and growth rates of $PM_{2.5}$ in Fig. 2? It seems directly relevant to the first scientific question.

**Response 6:**

Thank you for your helpful suggestion. The diurnal variations of the $PM_{2.5}$ and the corresponding analysis have been added. More details are as follows:

Notable differences are present when we compare the dilution-related parameters to $PM_{2.5}$. The daily maximum $PM_{2.5}$ concentrations in the spring, summer, autumn and winter were 73 μg m$^{-3}$ (11:00 LT), 56 μg m$^{-3}$ (09:00 LT), 78 μg m$^{-3}$ (23:00 LT) and 101 μg m$^{-3}$ (01:00 LT), respectively. The differences between the maximum and minimum were 14 μg m$^{-3}$, 10 μg m$^{-3}$, 20 μg m$^{-3}$ and 38 μg m$^{-3}$, respectively. Thus, the diurnal variation of $PM_{2.5}$ can be divided into two categories: (1) the highest value occurs in the midday in the spring and summer and the overall change is small and (2) the highest value occurs during the night in the autumn and winter and differs greatly from the lowest value (Fig. 2). The main causes of air pollution are local emissions and regional transport. Thus, these results indicate that there is a greater local contribution in the autumn and winter and higher regional transport in the spring and summer.

[Figure]

Fig. 2 Diurnal variations and growth rates of the MLH (a), WS$_{ML}$ (b), VC (c) and $PM_{2.5}$ (d) in the spring, summer, autumn and winter in Beijing. Diurnal variations are represented by lines and scatters.

Growth rates are represented by columns, and only positive values are shown in the figure.

**Comment 7:**
In Fig.3, it is worth discussing higher frequency of high VC ($> 10^3$ $m^2$ $s^{-1}$) in winter, is it due to high wind speed associated with frontal passage?
**Response 7:**
Thank you for your helpful suggestion. We agree with you. In winter, when the Siberian High transits, strong northwest winds prevail in the Beijing area (Fig. 3), resulting the higher frequency of the VC in the range of 1000-2000 $m^2$ $s^{-1}$. We explained this point in section 3.1.1 of the revised draft.

[Figure]

Fig. 3 Diurnal variations in the mixing layer transport flux of $PM_{2.5}$ and transport direction during different seasons in Beijing.

**Comment 8:**
In Fig.4, it seems to me that the dominant southerly wind partly explains the positive correlation between wind speed and $PM_{2.5}$ in summer.
**Response 8:**
Thank you for your helpful suggestion. The southern wind generally appeared at 12:00-2:00 LT, and the high $PM_{2.5}$ concentration generally appeared at 6:00-13:00 LT; therefore, there was no significant relationship between the two. In addition, due to the improper discussion of this section in the original text, we have deleted this section to avoid confusion.

**Comment 9:**
I don't think the conclusion on lines 289-294 that southerly wind is "dirtier" directly comes from Figure 5 and 6 Flux variation comes from $PM_{2.5}$ and wind speed, it could be that southerly wind are generally stronger. In order to demonstrate this point, it will help to add $PM_{2.5}$ fields in Figure 5 and Figure 6. Another way to demonstrate this conclusion is to show wind rose and flux rose, and $PM_{2.5}$ composite in different wind directions.
**Response 9:**
Thank you for your helpful suggestion. According your suggestion, the diurnal variation of the $PM_{2.5}$ concentration and the wind radar were added, and we found that the level of the TF is determined by two factors, the WS and $PM_{2.5}$ concentration. In the spring, summer and autumn, the strong south wind prevails in the afternoon. As the south wind is often accompanied by a high $PM_{2.5}$ concentration (Fig. 4), the TF is high.
In the winter, the whole day is dominated by westerly and northerly winds. Although
the northerly winds are strong, the TF is not high due to the low $PM_{2.5}$ concentration.
Generally, a high WS means fast mixing, and the corresponding MLH is also high. At
this time, the TF is mainly controlled by the WS. When the WS is low, the mixing speed
is slow, and the MLH is low. At this time, the TF is mainly controlled by the $PM_{2.5}$
concentration. From the above analysis, it can be inferred that if the MLH and WS
gradually decrease with the worsening of the pollution, the mixing layer TF is
controlled by the WS first and then by the $PM_{2.5}$ concentration, and the maximum TF
may occur at a critical moment. This moment is neither the moment of the maximum
WS nor the moment of the maximum $PM_{2.5}$ concentration but should be somewhere in
between.

[Figure]

Fig. 4 The wind radar in different seasons in Beijing.

**Responses to the comments from Reviewer #3**

We would like to thank you for your comments and helpful suggestions. We revised
our manuscript according to these comments and suggestions.

**General Comments:**
The manuscript presents a good investigation by studying the transport flux of
particulate matter in the mixing layer over Beijing area, one of the heavily polluted
places in the country. The study employs ceilometer, Doppler wind radar, and other
meteorological measurement techniques to determine the transport flux in the region.
Overall, the manuscript constitutes a good research article with clear conclusions, high
quality figures, and great organization of the data. However, there seems to be a lot of
room for English language improvement.

**Specific Comments:**
**Comment 1:**
Line 26, define "fine particle" for its first appearance, e.g., $PM_{2.5}$ or something else.
**Response 1:**
Thank you for your helpful suggestion. The definition of "fine particle" has been added
to the paper.

**Comment 2:**

Line 31, recommend changing to "Transport mainly occurs between 14:00 and 18:00
LT".
**Response 2:**
Thank you for your helpful suggestion. The text has been revised accordingly.

**Comment 3:**
Line 41, recommend changing "other provinces and cities" to "surrounding provinces
and cities"
**Response 3:**
Thank you for your helpful suggestion. The text has been revised accordingly.

**Comment 4:**
Line 46, define fine particulate matter as $PM_{2.5}$ also if it is what the authors mean
**Response 4:**
Thank you for your helpful suggestion. The definition of "fine particle" has been added
to the paper.

**Comment 5:**
Line 49, recommend changing "a steady decrease in poor air quality" to "steady
improvement in air quality"
**Response 5:**
Thank you for your helpful suggestion. The text has been revised accordingly.

**Comment 6:**
Line 77, recommend changing "...1.2% yr-1..." to "1.2 percent per year"
**Response 6:**
Thank you for your helpful suggestion. The text has been revised accordingly.

**Comment 7:**
Line 86, recommend changing "...the reliability of the model will decrease" to "...the
reliability of the model cannot be guaranteed"
**Response 7:**
Thank you for your helpful suggestion. The text has been revised accordingly.

**Comment 8:**
Line 91, recommend organizing it as "...transport flux (TF) in the mixing layer..."
**Response 8:**
Thank you for your helpful suggestion. The text has been revised accordingly.

**Comment 9:**
Line 156-158, the way this sentence and next one were constructed will really confuse
the readers. "Seasonal variation" means and focuses on the variation, i.e, the standard
deviation. I think the authors is trying to express something like this: "In terms of
seasonal variation, the means of MLH for spring and summer are relatively higher than those of fall/autumn and winter. However, WS was quite different from MLH, ...". For Line 166-169, according adjustment is recommended for the discussion of $PM_{2.5}$ to avoid confusion.

**Response 9:**

Thank you for your helpful suggestion. We apologize for this mistake. Similar errors in the full text have been corrected accordingly.

**Comment 10:**

Line 163-164, recommend changing to "...The average TC for summer, winter, and autumn were quite similar, with the VC values...."

**Response 10:**

Thank you for your helpful suggestion. The text has been revised accordingly.

**Comment 11:**

Line 233-234, does the authors want to express this: "When MLH, $WS_{ML}$ and VC were lower than 400 m, 2.5 m s$^{-1}$ and 1500 m$^2$ s$^{-1}$, respectively, the $PM_{2.5}$ concentration decline sharply with these parameters increasing"? It is hard to imagine air pollution declines at these conditions not in favor of atmospheric dispersion.

**Response 11:**

This section has been deleted. Thank you for your helpful suggestion, and we apologize for this mistake.

**Comment 12:**

Line 261, I think May TF of 269 mg m$^{-1}$ s$^{-1}$ was 1.5 times higher than August TF of 106 mg m$^{-1}$ s$^{-1}$. Alternatively, you can express it as "May TF was 2.5 times of August TF".

**Response 12:**

Thank you for your helpful suggestion. The text has been revised accordingly.

**Comment 13:**

A general comment: when using "transport" and "transportation", try to clarify it and avoid the ambiguity by meaning the transportation sector like vehicle emissions, since it is also great contributing factor for fine particle concentration.

**Response 13:**

Thank you for your helpful suggestion. Some ambiguity has been eliminated through the revision process, while the other instances can be understood by the context.

**Comment 14:**

Line 361-364, the expression in this segment could be revised to avoid negative image of the conclusion.

**Response 14:**

Thank you for your helpful suggestion. To avoid a negative image of the conclusion, this expression has been removed.

**Technical corrections:**
**Comment 1:**
Line 20, change "atmospheric pollution" to "air pollution"
**Response 1:**
Thank you for your helpful suggestion. The text has been revised accordingly.

**Comment 2:**
Line 24, change "weakens" to "weaker" or make alternative grammar corrections
**Response 2:**
Thank you for your helpful suggestion. The text has been revised accordingly.

**Comment 3:**
Line 35, change "transportation influence" to "influence/impact of (air pollutants)
transport", otherwise it seems to mean the influence of transportation section like
vehicles
**Response 3:**
Thank you for your helpful suggestion. The text has been revised accordingly.

**Comment 4:**
Line 45, change "the Beijing's air quality" to "Beijing's air quality"
**Response 4:**
Thank you for your helpful suggestion. The text has been revised accordingly.

**Comment 5:**
Line 48, change "Although Beijing's government has been dedicated..." to "Although
Beijing government has dedicated..."
**Response 5:**
Thank you for your helpful suggestion. The text has been revised accordingly.

**Comment 6:**
Line 49-50, change "...ensure the continuous decline..." to "...ensure continuous
decline..." or "...ensure the continued decline..."
**Response 6:**
Thank you for your helpful suggestion. The text has been revised accordingly.

**Comment 7:**
Line 109, change "...More detail descriptions..." to "More detailed descriptions..."
**Response 7:**
Thank you for your helpful suggestion. The text has been revised accordingly.

**Comment 8:**
Line 116, change "...remote sensor method..." to "remote sensing method..."
**Response 8:**
Thank you for your helpful suggestion. The text has been revised accordingly.

**Comment 9:**

Line 120, change the long dash to short dash or change it to "to"

**Response 9:**

Thank you for your helpful suggestion. The text has been revised accordingly.

**Comment 10:**

Line 150, change "...we carried out continuously measured..." to "...we continuously measured..." or "we carried out continuous measurement of..."

**Response 10:**

Thank you for your helpful suggestion. The text has been revised accordingly.

**Comment 11:**

Line 184, change "stable" to "relatively smaller"

**Response 11:**

Thank you for your helpful suggestion. The text has been revised accordingly.

**Comment 12:**

Line 185, recommend changing to "which are 4 h later than the peak and trough of MLH..."

**Response 12:**

Thank you for your helpful suggestion. The text has been revised accordingly.

**Comment 13:**

Line 193, change "at the latest" to "later than other seasons". "At the latest" means something else like a deadline.

**Response 13:**

Thank you for your helpful suggestion. The text has been revised accordingly.

**Comment 14:**

Line 195, change "TC" to "VC" or change "VC" to "TC", so that the same parameter is compared, even though we VC is used to express the magnitude of TC.

**Response 14:**

Thank you for your helpful suggestion. After careful consideration, we think that "atmospheric transport capacity" is prone to ambiguity, so we changed "atmospheric transport capacity (TC)" to "atmospheric dilution capability".

**Comment 15:**

Line 236, change to "...than other seasons..."

**Response 15:**

Thank you for your helpful suggestion. The text has been revised accordingly.

**Comment 16:**

Line 243, change "indicator factors" indicators" or "indicating factors"

**Response 16:**

Thank you for your helpful suggestion. The text has been revised accordingly.

**Comment 17:**
Line 255-256, need improvement for this expression: "The northwesterly and westerly
directions were the main transport sources of the cold period in Beijing."
**Response 17:**
Thank you for your helpful suggestion. This phrase has been revised to "The transport
sources of the cold period in Beijing were predominantly from the northwesterly and
westerly directions."

**Comment 18:**
Line 257, change "increased" to "changed"
**Response 18:**
Thank you for your helpful suggestion. The text has been revised accordingly.

**Comment 19:**
Line 286, change "rules" to "patterns"
**Response 19:**
Thank you for your helpful suggestion. The text has been revised accordingly.

**Comment 20:**
Line 297, change "4" to "four", please refer to manuscript preparation guidance about
numbers.
**Response 20:**
Thank you for your helpful suggestion. We apologize for our carelessness. The text has
been revised accordingly.

**Comment 21:**
Line 299, recommend changing "and we must pay attention to local pollutant emission
control" to "and local pollutant emission control is the most effective way of mitigating
pollution levels"
**Response 21:**
Thank you for your helpful suggestion. The text has been revised accordingly.

**Comment 22:**
Line 346-347, change "the concentration of pollutants has a good relationship with VC"
to "the concentration of pollutants is significantly correlated with VC"
**Response 22:**
This section has been removed. Thank you for your helpful suggestion.

Special thanks to you for your good comments.

We tried our best to improve the manuscript and made some changes accordingly.
These changes do not influence the content or framework of the paper. We did not list all the changes here, but they are shown in red in the revised manuscript. Furthermore,
to make the article more readable, we have had the manuscript professionally edited
for language.

We earnestly appreciate the Editor's/Reviewers' earnest work and hope that the
corrections will be met with approval.

Once again, thank you very much for your comments and suggestions.

Yours sincerely,

Dr. Tang

**References:**

[revised manuscript text omitted]

be used to inverse the vertical PM$_{2.5}$ concentration profile.
ThenAssuming that the
, the TFs in the mixing layer are calculated as
follows:

$$TF_u = \int_{i=1}^{n} (u_i - \frac{1}{n}\sum_{i=1}^{n} u_i \times C_{PM_{2.5}}C_i) \times \text{MLH}$$

$$TF_v = \int_{i=1}^{n} (v_i \times \mathrm{C}_i)\,TF_v = \frac{1}{n}\sum_{i=1}^{n} v_i \times C_{PM_{2.5}} \times MLH$$

(5)

Through the above method, radial and zonal TFes can be obtained, and vector synthesis in two directions can be conducted to obtain the main transport direction to find the transport source area.

**3. Results and discussion**

**3.1 Boundary layer meteorology**

**3.1.1 Seasonal variation**

To understand the variations of atmospheric dilution capability, we carried out continuous measured MLH and wind profile within the mixing layer over a 2-year period (2016.1.1-2017.12.31). The availability was verified after MLH elimination by Tang et al. (Tang et al. 2016). After the exclusion of the data of MLH under rainy, sandstorm and windy conditions, data availability was 95% over the 2-year period, higher than that of previous studies (Mues et al. 2017; Tang et al. 2016). The availability was lowest in February at 86% and highest in July at 99%.

In terms of  seasonal variation, the averages of  MLH for spring (781 ± 229 m) (value ± standard deviation) and summer (767 ± 219 m) were higher than those of  autumn (612 ± 166 m) and winter (584 ± 221 m) (Fig. 1). However, $WS_{ML}$ was  different from MLH in terms of seasonal variation, with the largest value at 4.6 ± 1.6 m s$^{-1}$ in spring, followed by winter (4.1 ± 2.7 m s$^{-1}$) and autumn (3.7 ± 1.6 m s$^{-1}$), and the smallest value at 3.6 ± 1.1 m s$^{-1}$ in summer. VC was calculated by the MLH and wind profile, and he results demonstrate that the dilution capability was strongest in spring, as the VC reached as high as 3940 ± 2110 m$^2$ s$^{-1}$. The dilution capability among summer, winter and autumn were similar, then the VC values were 2953 ± 1322 m$^2$ s$^{-1}$, 2913 ± 3323 m$^2$ s$^{-1}$ and 2580 ± 1601 m$^2$ s$^{-1}$, respectively. A monthly analysis shows that atmospheric dilution capability was  strongest in May, the VC was as high as 5161 ± 2085 m$^2$ s$^{-1}$,and worst in December, and the VC was only 1690 ± 1072 m$^2$ s$^{-1}$. The VC value in May was 3.1 times of that in December. To analyze the impacts of dilution capacity on $PM_{2.5}$, the seasonal variation of $PM_{2.5}$ were analyzed. The averages of $PM_{2.5}$ concentration for winter (80 ± 87 µg m$^{-3}$) was highest, followed by autumn (68 ± 54 µg m$^{-3}$) and spring (67 ± 60 µg m$^{-3}$), and summer (51 ± 29 µg m$^{-3}$) was lowest. The lowest monthly average $PM_{2.5}$ concentration was 42 ± 26 µg m$^{-3}$ in August. The highest monthly average was in January at 94 ± 100 µg m$^{-3}$, 2.2 times of that in August (Fig. 1).~~The seasonal variation in the PM2.5 concentration was the highest in winter (80 ± 87 µg m⁻³), followed by autumn (68 ± 54 µg m⁻³) and spring (67 ± 60 µg m⁻³), and the seasonal variation was the lowest in summer (51 ± 29 µg m⁻³). The lowest monthly average PM2.5 concentration was 42 ± 26 µg m⁻³ in August. The highest monthly average was in January at 94 ± 100 µg m⁻³, 2.2 times higher than that in August (Fig. 1). Thus, the vertical and horizontal diffusion capacities are strong in spring and weak in autumn and winter. In summer, the vertical diffusion capacity is strong, while the horizontal diffusion capacity~~

设置了格式: 字体: 倾斜
设置了格式: 字体: 倾斜
设置了格式: 字体: Times New Roman
设置了格式: 字体: Times New Roman
设置了格式: 字体: Times New Roman
设置了格式: 字体: Times New Roman
设置了格式: 字体: Times New Roman

[revised manuscript text omitted]
 high TF occurred. Therefore, the level of TF is determined by two factors, that of WS and $PM_{2.5}$ concentration. In spring, summer and autumn, strong south wind prevails in the afternoon. As the south wind is often accompanied by high $PM_{2.5}$ concentration (Fig. S2), TF is high. In winter, the whole day is dominated by westerly and northerly winds. Although the northerly winds are strong, TF is not high due to the low $PM_{2.5}$ concentration. Generally, high WS means fast mixing, and the corresponding MLH is also high. At this time, TF is mainly controlled by WS. While WS is low, the mixing speed is slow and MLH is low. At this time, TF is mainly controlled by $PM_{2.5}$ concentration. From the above analysis, it can be inferred that if MLH and WS gradually decrease with the worsen of pollution, the mixing layer TF is controlled by WS first and then by $PM_{2.5}$ concentration, which may appear a maximum TF at a critical moment. This moment is neither the moment of maximum WS nor the moment of maximum $PM_{2.5}$ concentration. It should be somewhere in between. This will be discussed in more detail in section 3.3.

[Figure]

Fig. 5 Diurnal variations in the mixing layer TF of $PM_{2.5}$ and transportation directions during different seasons in Beijing.

**3.2.2 Vertical evolution of TF**

After the aforementioned analyses, the transportation period is known. To further explore the height of transport, we studied the seasonal variation of TF profile in combination with the vertical wind and $PM_{2.5}$ profile. With the increase of altitude, the WS first increases sharply at approximately 200 m, and then slowly increases, and the differences between different seasons gradually become significant. WS is always smallest in summer, and strongest in winter. It is same in spring and autumn at 1200 m. Above 1200 m, WS in autumn exceeds those in spring. The $PM_{2.5}$ concentration at 100 m obtained by inversion is highest in winter (93.7 mg $m^{-2}s^{-1}$), and similar in spring and autumn (80.3 mg $m^{-2}s^{-1}$ and 75.8 mg $m^{-2}s^{-1}$, respectively), and lowest in summer (53.5 mg $m^{-2}s^{-1}$). This is consistent with near-surface results. Below 200 m, $PM_{2.5}$ concentration is relatively uniform. As the height increased, the $PM_{2.5}$ concentration decreased gradually. Between 200-600 m, $PM_{2.5}$ concentration begins to decrease rapidly, but the rate of decline was obviously different in different seasons. In autumn and winter, the reduce rate of $PM_{2.5}$ concentration was significantly higher than that in spring and summer. As a result, the spring $PM_{2.5}$ concentration at 400 m began to be greater than that in winter; the summer $PM_{2.5}$ concentration at 650 m began to be greater than that in autumn, and was at the same level as that in winter. Over 600 m, there is no significant difference in $PM_{2.5}$ concentration between different seasons, while WS varies greatly. Therefore, TF is greatly affected by WS at high altitude, while it is greatly influenced by $PM_{2.5}$ concentration on the near ground. Besides, the TF in the mixing layer is also affected by MLH.

The vertical evolution of the TF is different from both WS and $PM_{2.5}$ concentration, and the seasonal variation remains consistent from near-surface to the upper air, which shows that the TF for spring was the highest, followed by winter and autumn, and summer was the lowest. The vertical variation of TF increases firstly and then decreases, and a peak appears around 300 m, at 0.38 mg $m^{-2}s^{-1}$ in spring, at 0.19 mg $m^{-2}s^{-1}$ in summer, at 0.24 mg $m^{-2}s^{-1}$ in autumn, and at 0.31 mg $m^{-2}s^{-1}$ in winter. In the process of TF lowering, it has different performances in different seasons. In spring, the decline slowed down at about 1500 m. The change in summer and autumn is similar. After the peak, TF drops rapidly in summer and autumn. And the decrease rate above 500 m becomes slow, and then slows down again after 1500 m, finally, the TF profiles tend to be vertical. In winter, TF declines rapidly, followed by fluctuations around 1000 m. The above results can preliminarily indicate that the transportation mainly occurs within 200-1500 m, which will be dissect in Sec. 3.3. To sum up, in autumn and winter, the high concentration of $PM_{2.5}$ is concentrated in near the ground, while the TF is not large, again indicating that local emission is the main source of $PM_{2.5}$ in autumn and winter; in spring, affected by high-altitude transportation, $PM_{2.5}$ concentration is high; in summer, both TF and $PM_{2.5}$ concentration are at the lowest level, 
[revised manuscript text omitted]